# Promising Protocol for In Vivo Experiments with Betulin

**DOI:** 10.3390/pharmaceutics17081051

**Published:** 2025-08-13

**Authors:** Pavel Šiman, Aleš Bezrouk, Alena Tichá, Hana Kozáková, Tomáš Hudcovic, Otto Kučera, Mohamed Niang

**Affiliations:** 1Independent Researcher, Librantická 157, CZ-50003 Hradec Králové, Czech Republic; 2Department of Medical Biophysics, Faculty of Medicine in Hradec Králové, Charles University, CZ-50003 Hradec Králové, Czech Republic; 3Department of Research and Development, University Hospital Hradec Králové, CZ-50005 Hradec Králové, Czech Republic; alena.ticha@fnhk.cz; 4Laboratory of Gnotobiology, Institute of Microbiology of the Czech Academy of Sciences, CZ-54922 Nový Hrádek, Czech Republichudcovic@biomed.cas.cz (T.H.); 5Department of Physiology, Faculty of Medicine in Hradec Králové, Charles University, CZ-50003 Hradec Králové, Czech Republic; kucerao@lfhk.cuni.cz; 6Independent Researcher, Škroupova 497, CZ-50002 Hradec Králové, Czech Republic

**Keywords:** betulin, triterpenoids, in vivo administration protocol, determination of betulin in tissues, anticancer effect

## Abstract

**Background/Objectives**: Betulin is a promising agent in many areas of medicine and is being investigated, particularly in the field of cancer. However, in in vivo experiments, its water insolubility becomes a significant obstacle. This study describes a promising method for the administration of betulin in in vivo experiments and the determination of betulin levels in organ samples. **Methods**: Betulin was first dissolved in ethanol, and this solution was then mixed with acylglycerols, followed by evaporation of the ethanol. Olive oil and food-grade lard were determined to be suitable lipids for noninvasive application per os. A method for processing the organs of experimental animals for betulin determination was developed. Determination of betulin levels in blood is also likely the only viable option for use in future clinical studies and practice. **Results**: The maximum amount of betulin usable (i.e., absorbable by organisms) in olive oil (10 mg/mL), suppository mass (6 mg/mL), food lard (4 mg/mL), and cocoa butter (2 mg/mL) carriers was found microscopically. A specific distribution of betulin concentration in the organs of experimental animals (Wistar rats) after a weekly diet containing betulin was discovered. The blood was shown to be particularly advantageous, as it allows continuous monitoring of betulin levels in the body. In these pilot experiments, a statistically significant (*p* < 0.001) synergistic effect of betulin on solid Ehrlich adenocarcinoma tumors was observed when betulin was combined with cytostatic Namitecan (NMRI mice). The high-purity betulin used in this study is very stable even under fluctuating storage conditions. **Conclusions**: Our study suggests that both the method of betulin administration and the proposed analytical procedure could greatly increase the reliability and reproducibility of in vivo studies, as well as future preclinical and clinical studies on the effects of betulin and potentially other similar water-insoluble triterpenoids on living organisms.

## 1. Introduction

Betulin, a triterpenoid found in considerable amounts in birch bark (Betulaceae), is a promising agent in many areas of medicine and is being investigated, particularly in the field of cancer [1,2,3,4]. Betulin has been extensively studied in in vitro experiments, where its near insolubility in water is not a major drawback because of the possibility of using excipients, usually dimethyl sulfoxide (DMSO) [5,6]. However, in in vivo experiments, its insolubility becomes a significant obstacle, as it cannot be administered with confidence that it is sufficiently available in the body in molecular form to reach therapeutically effective levels. Many attempts have been made to deliver betulin into the bodies of experimental animals, including chemical modifications of betulin, but their results are controversial, e.g., Kuznetsova et al. [7] presented lethal doses for some animals that were quite unlikely to reach such levels in the animals’ bodies. Betulin and its numerous chemical derivatives [4], such as betulinic acid, lupeol, or other triterpenoids, are administered to experimental animals in a variety of ways. This is illustrated in detail in the work of Saneja et al. [8], who described examples of the administration of betulinic acid in the form of nanoemulsions, liposomes, polymeric or magnetic nanoparticles or polymer conjugates with carriers such as cyclodextrins, polyethylene glycols, biodegradable polymers, or even carbon nanotubes. A comprehensive review of the administration of betulin and its chemical derivatives by Jaroszewski et al. [9] reported similar strategies, with an emphasis on nanoparticulate drug delivery using organic or inorganic nanocarriers. While these systems are promising, they are relatively difficult, expensive, and/or do not provide adequate or controllable delivery of the selected triterpenoid to the tissues. They often suffer from limited reproducibility, batch-to-batch variability, or uncertain pharmacokinetics. In contrast, our study presents a simple and reproducible protocol for achieving true molecular solutions of betulin suitable for in vivo use, which may serve as a practical alternative for preclinical research.

The routes of administration also vary considerably, especially in terms of their efficacy and invasiveness: oral, intravenous, subcutaneous, intramuscular, intraperitoneal, or inhaled. Many studies have not even described the exact procedure used to prepare the form of betulin; sometimes, the relevant citation is not even given but only mentions that a certain amount of betulin was administered orally [10,11], intraperitoneally [12], etc. In such cases, betulin is usually used in the form of a suspension of varying fineness (such as by Zakrzeska et al. [13]) and not in the form of a true solution. The bioavailability of this form of betulin is very controversial and usually very low [4,9,14,15], and good reproducibility cannot be assumed. Greater bioavailability, but only in extrinsic use, can be achieved by using a betulin solution in Oleogel-S10 (a mixture of oils and glyceryl behenate [16,17]); however, even here, the direct dissolution of betulin is chosen, which does not ensure the perfect molecularity of the solution.

The aim of this study was to develop the simplest noninvasive way of using betulin in experimental animals with respect to its possible future simple, inexpensive, and reproducible use in medicine. Moreover, we designed a simple method of processing animal tissues to determine the concentration of betulin in biological samples, i.e., to determine the effective concentration in the organism.

## 2. Materials and Methods

### 2.1. Materials and Equipment

#### 2.1.1. Animals

NMRI mice (35 males used for the final experiments that provided the resulting data +15 males used for the optimization of individual methodologies (e.g., administration processes, experiments with clotted blood) without relevant experimental data, weighing 32–40 g) and Wistar rats (2 males, weighing 233 and 245 g) were all fed standard laboratory mouse/rat chow (Velaz, Czech Republic) and water ad libitum, under laboratory conditions, on a 12:12 h light-dark cycle.

#### 2.1.2. Drugs and Chemicals

Lupeol (>94%, Sigma-Aldrich, St. Luis, MO, USA), Namitecan (Sigma-Aldrich, St. Louis, MO, USA), olive oil (pharmaceutical grade), suppository mass (pharmaceutical grade), food-grade lard, food-grade cocoa butter, ethanol (96%, VWR, Fontenai-sous-Bois, France), NaOH (Sigma-Aldrich), and chloroform (Sigma-Aldrich) were used in this study.

Betulin (99.7%) was prepared according to the improved method of Šiman et al. [5] (Figure 1). The ethanol extract from outer birch bark dried in 96% ethanol was dried. Then, the extract was dissolved in hot acetone (>40 °C), triterpenoids were precipitated with water, and the precipitate was filtered and dried (after this step, all the products were crystalline, and for subsequent handling with precipitates, only filtration was sufficient). The precipitate was dissolved in hot ethanol (>40 °C) and betulinic acid, and other acidic substances, polyphenols, and lipids were removed from the hot solution by adding an ethanolic solution of CaCl_2_ and then an equimolar ethanolic solution of NaOH during intensive stirring. Impurities were removed from the precipitate with Ca(OH)_2_ and NaCl via filtration. The filtrate was concentrated by evaporation up to 1/3–1/4 of the original volume, and crude betulin and lupeol were freely crystallized (~4 °C). Dried crystals were dissolved in chloroform (~1 part of the crystals was dissolved in 16 parts of chloroform (*w*/*v*)), and betulin was precipitated by the addition of petroleum ether (lupeol is much more soluble in hydrocarbons than betulin), followed by filtration of the precipitate and washing with petroleum ether. This step was repeated twice. Dried precipitate was dissolved in chloroform and filtered through silica gel (1–2 cm high column), and the filtrate was dried (to yield a snow-white amorphous substance). Pure betulin was crystallized from a hot ethanolic solution via cooling. The melting point was determined in the previous work by Siman et al. [5] (255–256 °C).

#### 2.1.3. Chromatography

Gas chromatography with mass spectrometry (GC-MS; Agilent Technologies—GC 7890A, MS 7890A, USA) detection was chosen for quantification of the isolated triterpenes (Šiman et al. [5], Caligiani et al. [18], Hassan et al. [19]). The use of the GC-MS method was given by available equipment and established practices, but liquid chromatography methods can be significantly more sensitive (e.g., UHPLC-MS; Rathod et al. [20]). The estimated sensitivity of the GC-MS used in our experiment was significantly below 1 μg/g of the sample.

#### 2.1.4. Microscopy

Microscopically, the maximal concentration of betulin in the carriers was determined via a Nikon Eclipse 90i microscope with a Plan Apo 40x DIC M N2 (Nikon Corporation, Tokyo, Japan) and NIS-Elements AR 3.20 software (Laboratory Imaging, Prague, Czech Republic).

### 2.2. Methods

#### 2.2.1. Administration Form

Betulin was dissolved in hot ethanol (>60 °C) at a concentration of 20 g/L. This solution was then added in appropriate amounts in batches to the selected carriers (olive oil, suppository mass, food-grade lard, and food-grade cocoa butter) at 80 °C, and the mixture was further heated with stirring until the ethanol was completely evaporated. Attempts were also made to add the entire amount of ethanol solution at one time, with virtually the same result. Since the hot ethanol solution evaporates rapidly in air, care must be taken during addition so that the nascent betulin microcrystals on the solution container do not contaminate the carrier. However, slight foaming of the carrier after the addition of the ethanol solution is not harmful.

To determine the concentration of betulin that can be used in the carriers, the resulting concentrations of 2, 4, 6, 10, and 16 mg of betulin per g of carrier were chosen. The melted samples (60–80 °C) were assessed microscopically via enriched carriers shortly after preparation and then after storage at 4 °C for three weeks. All images were taken under the same illumination conditions. The resulting focused image was created with a set of successive slices with a step size of 1.2 µm on the Z axis.

#### 2.2.2. Administration of Betulin per Os

The rats were fed a standard diet supplemented with lard containing betulin. The powdered diet was mixed with lard prepared according to the procedure described above with a betulin concentration of 3 g/kg of lard to obtain a final dietary betulin concentration of 150 mg/kg (60 g lard + 1140 g standard diet). The pellets were prepared from the mixture and made available to the animals ad libitum for 1 week. The animals were then sacrificed, and samples of heparinized blood, spleen, liver, kidney, heart, and adipose tissue were collected to determine the betulin content via the procedure described below.

Twenty-eight mice used in the Namitecan–betulin experiment (see Section 3.5) were fed a standard diet, and betulin was applied by gavage as a solution in olive oil at a dose of 5 μg of betulin per 1 g of body weight on days 2, 3, 5, and 7 after tumor transplantation. The enriched olive oil was prepared via the method described above at a concentration of 0.5 mg per 1 mL of oil.

#### 2.2.3. Administration of Betulin per Rectum

After overnight starvation, 7 mice were injected with 200 μL of a liquid suppository (37 °C) containing betulin (6 mg/mL) via a catheter per rectum. After 2 and 5 h, the mice were sacrificed, and betulin levels in the blood, liver, spleen, jejunum, and duodenum were determined.

#### 2.2.4. Determination of Betulin Concentrations in Organs

Heparinized blood was separated into plasma and cell sediment by centrifugation, and solid organs were roughly homogenized in 1 mL of saline. The amount of processed solid samples did not exceed 1 g, and 0.5 mL of plasma (with 1 mL of saline) was always used. All the samples were further processed in glass test tubes to avoid the adsorption of hydrophobic betulin or lupeol at the surface. To all the samples, 4 μg of lupeol in 50 μL of ethanolic solution (200 mg/L) was added as an external standard. Then, 2 mL of a 20% NaOH solution was added, and the mixture was shaken at a temperature of approximately 60 °C until complete dissolution of the tissue was achieved, as reflected by the total transparency of the solution. After the addition of 2 mL of chloroform, the mixture of the two immiscible liquids was shaken vigorously at normal temperature, and each sample was shaken for a total of approximately 5 min. The two liquid phases were then separated by centrifugation (1500× *g*/10 min), the lighter aqueous phase was carefully removed, and 1 mL of the pure chloroform phase was collected in vials. The vials were left open overnight at laboratory temperature in a fume hood to allow the chloroform to evaporate completely. The samples were further processed and measured via gas chromatography with mass spectrometry detection. Standardization was performed in the same manner using 0.5 mL of intact blood, to which lupeol (4 μg) and betulin (2, 5, 10, 20, 50, and 100 μg) were added as standards for constructing a calibration curve.

Lupeol is used as an internal standard for GC-MS but is considered an external standard from a biological point of view. It was selected for its structural similarity to betulin and its chromatographic behavior. However, it is not ideal for pharmacokinetic use due to its own biological activity.

#### 2.2.5. Synergistic Effects of Namitecan and Betulin

Twenty-eight mice were subcutaneously injected with a solid form of Ehrlich’s adenocarcinoma, resulting in a total of 10^6^ cells. The mice were randomly divided into 4 groups of 7 animals each: 1. control without betulin and Namitecan; 2. with betulin only; 3. with Namitecan only; and 4. with both betulin and Namitecan. The cytostatic drug Namitecan was injected on days 3 and 7 after tumor transplantation (15 μg/g of body weight) [21], and betulin was applied per os by gavage on days 2, 3, 5, and 7 after tumor transplantation, as described in Section 2.2.2. The groups without betulin (1st and 3rd) received an equivalent amount of olive oil without betulin by gavage at the same time. The mice were then left without further treatment for the survival test. The duration of the experiment was 36 days, which corresponds to the survival time of the longest living mouse.

#### 2.2.6. Statistical Analysis

The data were statistically evaluated with NCSS 10 statistical software (2015, NCSS, LLC., Kaysville, UT, USA) and MS Excel 2016 (Microsoft Corp., Redmond WA, USA). The normality of the data distribution was tested by the kurtosis normality test and was not rejected. Therefore, the data are presented as the mean and standard deviation of the sample ($\bar{X}$ ± SD). The solid Ehrlich tumor weights after different types of treatment were compared via the 2-sample *t* test (and Levene’s test was used to check the homoscedasticity assumption). A *p* value of less than 0.05 was considered statistically significant.

## 3. Results

### 3.1. Administration Form

It was proven microscopically that by introducing an ethanol solution of betulin into heated acylglycerols, clear solutions without microscopically visible microcrystals of betulin were obtained. Only after a certain concentration was reached could easily recognizable long microcrystals or characteristic “hedgehog-like” crystal clusters with a size of μm begin to fall out. Among the carriers studied, olive oil was the most suitable. It remained as a clear solution up to a betulin concentration of 10 mg/mL. Suppository mass could absorb up to 6 mg/mL without precipitating crystals, and food lard could absorb at least 4 mg/mL. Cocoa butter proved unsuitable, as betulin microcrystals appeared even at a concentration of 2 mg/mL. Storage at 4 °C for one month revealed that the solutions were relatively stable; after repeated melting by heating, the samples were similar. Figure 2 shows sections of microphotographs with margins of approximately 150 μm from three typical examples.

### 3.2. Administration of Betulin per Os

Betulin administration using lard in feed seemed problem-free. Standard feed enriched in lard tasted good to the rats. Unfortunately, we did not measure the consumption of enriched feed in comparison with standard feed.

Both methods of betulin administration, with lard in feed and via gavage as a solution in olive oil, were successful. This finding was confirmed by measuring betulin levels in the organs of the rats and the visible effect on the mass of the solid tumors in the mice together with Namitecan, as shown below.

The method used indicated an order of magnitude greater availability and two–three orders of magnitude greater efficiency (considering the amount of betulin administered) of the given form of betulin solution than the use of conventional suspensions for per os or intraperitoneal administration [14,15].

### 3.3. Administration of Betulin per Rectum

The administration of betulin per rectum using a suppository mass as a carrier did not provide the desired results. Although some detectable amounts of betulin appeared in the jejunum 5 h after application, the level in whole blood remained below the detection limit of the method.

### 3.4. Determination of Betulin Concentrations in Organs

The method for determining betulin in organs is simple to perform and sensitive enough to reliably determine therapeutically appropriate amounts of betulin (e.g., 5 μg/g according to the IC50 in the case of the triple-negative breast tumor cell line BT-549-Šiman et al. [5] or, in the case of some other cell lines, Król et al. [1]). The authors estimate the sensitivity limit to be much less than 1 μg/g of tissue when GC-MS is used. Heparinized blood appears to be the best among all organs tested.

Only two rats were used to determine the distribution of betulin administered per os. Therefore, the “statistics” represented by the standard deviation in the following figure are highly questionable. Nevertheless, notably, there were very few differences in all the organs studied between the two rats, with the exception of adipose tissue. Thus, it appears that the two animals were fed identically with respect to their weight and that the equilibrium distribution of ingested betulin in the organs was quite characteristic. The adipose tissue undoubtedly contained a considerable amount of betulin, but processing of the samples by the method described was unsuitable for this tissue. The high fat content led to the formation of large amounts of sodium soap, which precipitated as a shapeless mass containing an unknown amount of betulin not available for extraction to chloroform. For adipose tissue, therefore, the methodology would have to be modified.

Figure 3 shows substantial differences among various organs (up to sevenfold between kidney and blood sediment), especially compared with small individual “standard deviations”. These results were obtained by the GC-MS method.

### 3.5. Synergistic Effects of Namitecan and Betulin

The results of this pilot study on the effect of betulin in cancer treatment are promising. Although betulin alone is not effective (*p* = 0.0361) in the in vivo treatment of Ehrlich’s adenocarcinoma (Figure 4), as reported in some tumor types even in vitro [1], it can act synergistically with other cytostatic drugs. The cytostatic drug Namitecan showed, as expected, high treatment efficiency (*p* < 0.001). However, the combination of Namitecan with betulin showed synergistic effects, and an even greater efficiency was achieved in the in vivo treatment of Ehrlich’s adenocarcinoma than in the control (*p* < 0.001) and Namitecan alone (*p* = 0.024) (Figure 4).

## 4. Discussion

### 4.1. Carriers with Betulin

The acylglycerols used were relatively good carriers of molecular betulin. In particular, olive oil allows the uptake of therapeutic amounts of betulin by organisms. The lard can also be used as part of a mixed diet (but it is not well defined as a carrier), since the rats seemed to like an enriched diet. However, it would be extremely difficult to obtain clear solutions by merely dispersing pure betulin in given carriers, since crystalline betulin dissolves very slowly even in hot ethanol, for example. The term “clear solutions” used in the article regarding betulin solutions in carriers means that the authors cannot be sure that they are true monomolecular solutions. Indeed, the structural formula of betulin suggests that in these solutions betulin may be present in dimers where two molecules bind to each other by hydrogen bonding of their -OH groups. However, the authors do not assume the presence of permanent larger clusters, since, especially at the high concentrations of betulin that are applicable in olive oil (min. 10 mg/mL), the growth of such clusters would progress to the stage of microscopically visible microcrystals. What is more important, however, is that betulin in these clear solutions appears to be readily bioavailable to the organs of the experimental animals.

The evaporation of ethanol from the carriers could be further improved by reducing the pressure while non-reactive gas (e.g., nitrogen, argon) bubbles pass through. The carriers used were not pretreated in any way. However, it is conceivable that they could largely absorb betulin after thorough drying, for example. For such drying, non-reactive gas (e.g., nitrogen, argon) bubbling through a warm liquid carrier up to 100 °C with reduced pressure could be suitable for the pharmaceutical production of cachets with betulin-enriched olive oil.

The use of a simple ethanol solution of betulin placed directly on food, i.e., without the use of a proper carrier [22], is highly questionable. This approach is based on the authors’ experience with betulin solutions in various solvents. The ethanol evaporates, and betulin precipitates as microcrystals. Thus, the availability of betulin to experimental animals is extremely limited, and it would be difficult to detect betulin in experimental subjects and achieve reproducible results.

The proposed method of administration may lead to revision of some quantitative data from in vivo experiments, especially in the case of pharmacokinetics [4] and safety or lethal doses of triterpenoids [7].

Compared to nanoformulations, the presented protocol does not require surfactants or specialized equipment and enables consistent preparation of clear solutions. This may be advantageous in preclinical settings, especially where simple and reproducible drug delivery is preferred.

### 4.2. Determination of Betulin Concentrations in Organs

Most in vivo studies avoid determining the levels of betulin in the organism, as such determination places high demands on objectivity. The method described in Jäger et al. 2008 [14], for example, is controversial, and it is difficult to deduce the real betulin level from it, as the following lines of discussion show.

For the determination of betulin in tissues, it is necessary not only to dissolve the tissue but also to saponify fats and hydrolyze all polymers with hydrophobic components, e.g., proteins. Only then can it be assumed that all betulin is available for chloroform extraction. In addition, the extraction of original fats and fatty acids can decrease the efficiency and reproducibility of the final chromatographic determination. The solution to the aforementioned problems is to use a high concentration of NaOH, which hydrolyzes most of the polymers presented and converts the fats and fatty acids into the ionic form of soaps. Both betulin and lupeol are stable even at this high pH. Since extraction with chloroform is carried out at laboratory temperature and over a relatively short period of time, there is practically no risk of a noticeable reaction between chloroform and NaOH.

However, the method used for the hydrolysis of polymers and saponification of fats with high NaOH concentrations is unlikely to be suitable for accessing acidic triterpenoids such as betulinic acid, oleanolic acid, or ursolic acid. Here, it is virtually certain that they are ionized at high pH and cannot be quantitatively extracted into chloroform.

Only two rats were used for the determination of betulin in the organs; therefore, no statistically proven conclusions could be drawn. However, the very small variations in the values obtained, with the exception of adipose tissue, where massive precipitation of soaps was probably the main cause, suggest at least sufficient distribution of betulin administered per os among the different organs after one week. This finding also suggests which organ tumors sensitive to betulin are most affected. Given that the very small deviations between the organs of the two rats can hardly be the work of a happy accident, the results also demonstrate the high reliability of the proposed method for determining betulin in organs.

Since there are no studies on the basis of which the appropriate time of administration could be determined, this was determined by estimation to establish an equilibrium concentration of betulin in the rats. Importantly, this study is the first pilot study to create an effective form of betulin administration, in which betulin remains safe in the molecular form of the clear solution and is therefore absorbable by the organism. This study is also the first to provide a method for determining true betulin levels in organs. Only now will it be possible to carry out further necessary experiments, e.g., pharmacokinetics of betulin, histological control of basic vital organs, and monitoring of the overall condition of the animals after longer-term exposure to betulin, which will enable a qualified determination of the appropriate administration time.

The determination of betulin in complete heparinized blood could reflect the level of biologically available betulin in the body. On the one hand, there is a significant amount of betulin in the blood, and on the other hand, the blood can be taken easily during the entire period of betulin administration. For example, pharmacokinetic studies can be performed relatively easily. In clinical trials, this is also the only easy way to monitor betulin levels in the body during treatment. Blood lipoproteins are likely the main carriers of betulin in the body.

An attempt has also been made to determine betulin in clotted blood, but this has not been successful. Initially, it was quite difficult to dissolve blood clots. Such samples were then often strongly discolored in the chloroform layer and thus had to be removed from samples for measurement by GC-MS. Therefore, it is far better to use blood that has not clotted (e.g., heparinized).

Fifteen males of NMRI mice were used for the optimization of individual methodologies, e.g., administration processes and experiments with clotted blood, without relevant experimental data. However, important information emerged from these experiments, such as the inappropriateness of using clotted blood to determine the content of betulin in the body.

Although a full validation of the analytical method—including parameters such as extraction recovery, matrix effect, accuracy, and precision—was originally planned, it could not be completed due to the loss of original data and departure of key contributors. Therefore, the method presented in this study should be considered preliminary. Nevertheless, it offers sufficient sensitivity and robustness for further development and use in pharmacokinetic studies.

As pharmacokinetic profiling was not part of the current pilot experiment, no plasma time-course data are available. However, the described methodology enables such experiments, and we recommend that future studies investigate absorption kinetics, distribution, metabolism, and elimination of betulin using our proposed analytical approach.

Originally, tissue distribution was performed on six rats. Unfortunately, after the loss of experimental data linked to the passing of a key co-author, only two datasets remained available and usable. We are fully aware that such a low number of animals does not allow for statistical analysis, and we present the data in Figure 3 only as a qualitative illustration of betulin distribution in major organs.

The determination of betulin in the organs of the mice was not planned in this study because of the use of a very low concentration of betulin in the applied olive oil and the resulting assumption of the probable disappearance of betulin in the organs of the mice at a time shorter than their expected survival time.

### 4.3. Synergistic Effects with Namitecan

The results of this pilot study on the synergistic effect of betulin with other cytostatic drugs in tumors insensitive to betulin alone are promising. The synergistic effect of using Namitecan with betulin in the in vivo treatment of Ehrlich adenocarcinoma was demonstrated, as the group treated with Namitecan with betulin presented significantly lower Ehrlich tumor weights than the group treated with Namitecan alone. The difference in survival times was not statistically significant in this pilot experiment; however, the results indicated a positive trend (Figure 4). It is very likely that very good results could be obtained after appropriate modification of the experimental protocol.

Notably, only a very small amount of betulin was used, and its concentration in the administered olive oil could be increased up to 20-fold. Betulin could also be administered for a longer period than it was in the experiment. All this could have significantly changed the survival of the experimental animals. In the case of strong statistical evidence of the synergistic effect in subsequent full-scale experiments, the use of betulin may reduce the doses of cytostatic drugs, reducing their side effects.

Moreover, betulin administered together with a more aggressive cytostatic drug may also have a protective effect on organs, as suggested, for example, in Lee et al. [23], in which triterpenes showed a renoprotective effect when cis-platinum was used.

Although the current study demonstrated a statistically significant synergistic effect between betulin and Namitecan, we were unable to collect pharmacokinetic or tissue distribution data in tumor-bearing mice due to the unexpected loss of experimental capacity. These experiments were originally planned and would have significantly strengthened the findings. Future studies should include detailed PK analysis and distribution profiling under combination treatment conditions. Although no toxicity was observed clinically during the experiment, no formal biochemical or histopathological evaluation was performed. This remains a key limitation of the current study and a priority for future research.

The exact mechanism behind the observed synergy between betulin and Namitecan remains unknown. However, based on previously published studies, several potential pathways could be involved. Betulin has been reported to induce apoptosis and modulate oxidative stress in melanoma cells [5,24]. Namitecan, a camptothecin analog, inhibits topoisomerase I and mediates DNA damage-induced apoptosis [25]. Their combined effect may therefore result from complementary actions on apoptotic signaling and enhanced cellular stress. Further mechanistic studies using molecular and cellular approaches are needed to clarify this interaction.

### 4.4. Stability of Pure Betulin and Its Formulation

The crystalline form and purity of highly pure betulin did not change, as detected microscopically (Figure 5) and by GC-MS, even after five years of storage in the dark in a well-closed container and at temperatures ranging from approximately 15 to 40 °C. The recommended storage times, especially the expiry times, of commercial products are therefore often excessively strict. However, this may also be related to purity, as the betulin used was extremely pure (99.7%) according to GC-MS analysis.

Chemically, pure crystalline betulin is extremely stable, as confirmed by our own long-term storage observations (no detectable degradation after five years). Therefore, degradation of the active substance is highly unlikely under standard storage conditions. However, the physical stability of the lipid-based formulation—particularly recrystallization or phase separation—may be affected by temperature fluctuations over time. Although not systematically evaluated in this study, this aspect should be considered in future formulation development and long-term stability studies (e.g., at ambient, refrigerated, and accelerated conditions).

In addition, chemical stability under varying pH and light exposure conditions was not evaluated in this study. These factors may influence degradation kinetics in clinical formulations and should be systematically examined in future development.

### 4.5. Microbial Stability

Microbial stability of betulin is sufficient, as the crystalline product obtained by final ethanol crystallization is sterile and can be stored under sterile conditions. During formulation, it is reintroduced as a hot ethanol solution, allowing additional thermal sterilization. The literature also reports on the antimicrobial, antiviral, and antifungal activity of betulin itself, which may further reduce the risk of microbial contamination.

### 4.6. Strengths and Limitations

The method of preparing betulin for per os application enables the generation of a clear solution of betulin in the selected carrier with a completely reproducible composition and reproducible bioavailability for the organism under investigation. This will now allow further in vivo experiments to be evaluated truly quantitatively, not just qualitatively in nature. The second method, the analysis of samples, enables continuous and reproducible determination of the actual levels of betulin in the organism under investigation.

All the preliminary observations mentioned above need to be confirmed by further research and on a larger scale. However, the promising and statistically significant results from this pilot small-scale experiment provide hope for further full-scale studies on the anticancer effects of betulin and for preclinical and clinical studies.

Thus, the authors believe that the present work can be the basis for research by other scientific groups with the ultimate goal of successful clinical trials using the natural substance betulin, an inexpensive drug with an almost infinite source in birch bark.

However, due to the unfortunate loss of key team members, full method validation (including recovery, matrix effect, accuracy, and precision) as well as pharmacokinetic profiling in plasma could not be completed. These limitations are explicitly acknowledged and clearly defined in this revised version. We believe that the presented method offers a useful foundation for other research groups who may wish to carry out these critical next steps. This also applies to pharmacokinetic and tissue distribution experiments in tumor-bearing animals under combination treatment, which were part of the original study plan.

## 5. Conclusions

The introduction of a hot ethanol solution of betulin to warm acylglycerols seems to be the best method to obtain a clear solution of betulin in carriers such as olive oil, suppository mass, or lard for per os administration of this triterpenoid in in vivo experiments. The concentration of betulin in such carriers may be sufficient for reaching therapeutic levels in experimental animals or, in the future, in the bodies of potential patients. This method of administration is gentle to animals or patients and is inexpensive, relatively quick, and effective with the delivery of betulin to the body. The suggested method of administration is supplemented by a relatively simple analytical method for the determination of betulin in organs, especially in the blood. The samples were dissolved in NaOH solution, and then betulin was extracted with chloroform. The content of betulin in the evaporated residue of the chloroform solution was then analyzed chromatographically.

The present method of administering betulin proved to be suitable for studying its synergistic action with the cytostatic drug Namitecan. Although this study represents the first pilot experiment, and the amount of available betulin was very small, the results proved to be promising, and the synergistic effect was statistically proven.

## Figures and Tables

**Figure 1 pharmaceutics-17-01051-f001:**
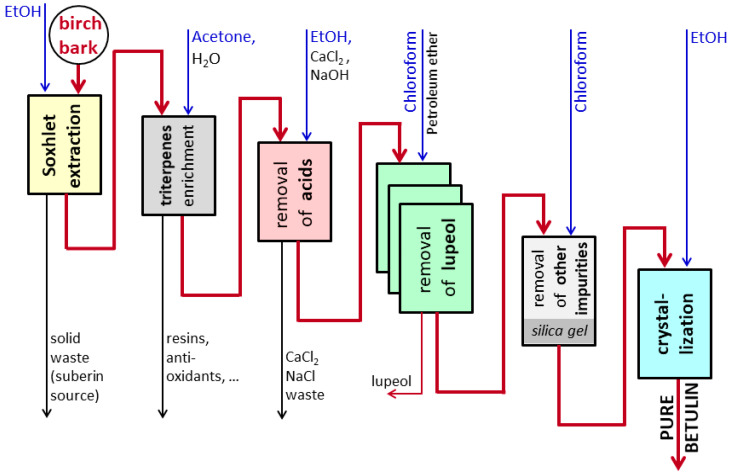
Scheme of pure betulin preparation (modified from Šiman et al. [5]).

**Figure 2 pharmaceutics-17-01051-f002:**
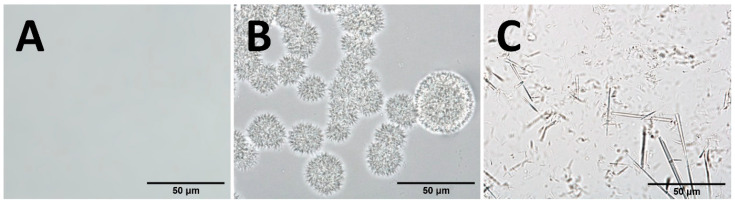
Photomicrographs of betulin in carriers: (**A**) olive oil (10 mg/mL); (**B**) olive oil (16 mg/mL); (**C**) cocoa butter (4 mg/mL). The selected photos were taken within a maximum of 6 h after the preparation of the solutions.

**Figure 3 pharmaceutics-17-01051-f003:**
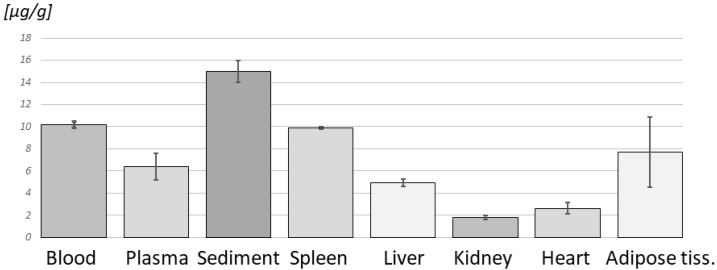
Betulin in rat tissues. Levels of betulin after one week of administration of lard with betulin (μg of betulin/g tissue; n = 2), obtained by the GC-MS method. Only two rats were available for final analysis due to partial data loss. Results should be interpreted as qualitative only.

**Figure 4 pharmaceutics-17-01051-f004:**
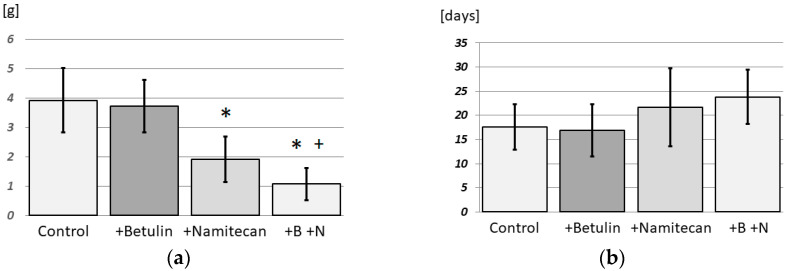
(**a**) Mass of solid Ehrlich’s tumor after treatment with betulin and/or Namitecan; (**b**) Namitecan and survival of treated mice. n = 7 in control, betulin, and Namitecan; n = 6 in N + B. The asterisk (*) denotes significantly lower Ehrlich’s tumor weight in comparison with the control group (+Namitecan; +B +N; *p* < 0.001). The plus sign (+) denotes significantly lower Ehrlich’s tumor weight than the Namitecan group (*p* = 0.024).

**Figure 5 pharmaceutics-17-01051-f005:**
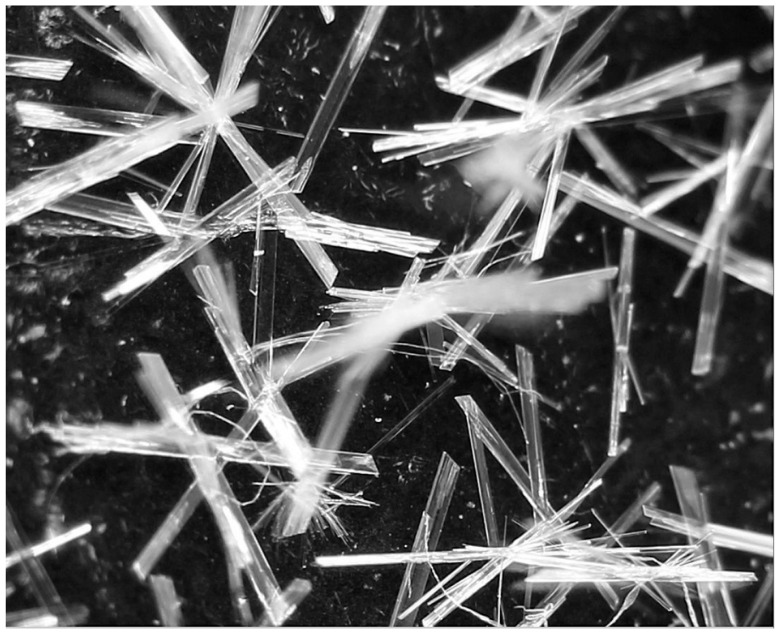
Pure betulin five years after preparation. Originally recrystallized from a nearly saturated ethanol solution at about 70 °C by cooling to about 4 °C. Photographed on the equipment mentioned in Section 2.1.4.

## Data Availability

The data that support the findings of this study are available from the corresponding author upon reasonable request.

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
