# Peer review of "Promising Protocol for In Vivo Experiments with Betulin"

_pharmaceutics, 2025, doi:10.3390/pharmaceutics17081051_

Round 1
Reviewer 1 Report (Previous Reviewer 3)
Comments and Suggestions for Authors
Please see the attachment.

Author Response
Dear Reviewer, we would like to thank you for your valuable review and comments, which – we believe – helped us significantly to improve our manuscript. Our amendments stemming from your requests are listed in the attached Word document.

Reviewer 2 Report (New Reviewer)
Comments and Suggestions for Authors
The work is well planned and executed. A shortcoming is the small number of samples analyzed, but this is understandable given the need for animal studies.
The first part of the paper presents a method for obtaining betulin preparation and an analytical method for determining betulin concentrations in organs, especially in blood, is not particularly innovative, but may be of interest to other researchers using betulin in their study.
The second part of the study, concerning the synergistic effect of betulin with the Namitecan, is much more interesting. Unfortunately, these studies are very preliminary and raise many questions that remain unanswered. First, why was Namitecan chosen as the cytostatic drug, which is not used as an anticancer drug? It would be probably better to conduct such studies with a drug that is already in clinical use. These studies are more innovative, but unfortunately, they are underdeveloped.
Despite some shortcomings and imperfections, I think the article is worth publishing because it contains in vivo studies and may be helpful and interesting for other researchers working on betulin.
The article requires corrections in section 2. Materials and methods.
1. Please enter the correct chapter numbering.
2. The title of section 2.1.2 is incorrect because it describes not only the drugs but also all the substances/reagents used.
3. Section 2.1.5 should be incorporated into 2.1.2 and its title changed to "Preparation of betulin" instead of "Brief description of the betulin preparation."
4. The description of betulin preparation should not be presented in bullet points, but as a single text.
Author Response
Dear Reviewer, we would like to thank you for your valuable review and comments, which – we believe – helped us significantly to improve our manuscript. Our amendments stemming from your requests are listed in the attached Word document.

Reviewer 3 Report (New Reviewer)
Comments and Suggestions for Authors
pharmaceutics-3749571
Comments and suggestions for authors
The manuscript entitled "Promising protocol for in vivo experiments with betulin” by Pavel Šiman, Aleš Bezrouk, Alena Tichá, Hana Kozáková, Tomáš Hudcovic, Otto Kučera, Mohamed Niang submitted to the section ‘Drug Delivery and Controlled Release’, represents an analytical study of the natural product betulin in rat blood and organs after oral administration.
Overall, the manuscript of this analytical study leaves a dual impression. Indeed, the use of natural biologically active compounds created by nature, in medicine is a modern trend. And the product contained in birch bark – betulin – has long been used as a disinfectant and anti-inflammatory agent. Currently, betulin is in high demand in pharmacological research. Therefore, the presented work is very relevant.
On the other hand, the purpose of this study is not clearly defined. And, as a result, the research as a whole is fragmentary and unfinished. Moreover, some important experiments, for example, on the distribution of betulin in organs, were conducted on a group of two animals (rats). This is unacceptable for scientific research.
At the same time, this study, of course, is interesting and versatile; it was carried out at a good scientific level. The authors found a simple method for the quantitative determination of triterpenoids demonstrated adequate validation parameters.
It is written in good English and easy to read. All cited literature is well structured. The represented results are of interest, first of all, for the pharmaceutical, analytical, and chemical communities.
I recommend its publication in Pharmaceutics after major revisions.
Regarding the manuscript I have a number of comments:
1) Title. The title of the article weakly reflects its content.
2) Materials and Methods. Section 2.1.5. The pure product crystallized from hot ethanol (step 6) should be characterized by modern methods, such as NMR, FTIR, etc. Finally, the melting point of betulin should be determined.
3) Results. Information on GC-MS results must be provided.
4) Figure 5. Give more details: what solvent were the crystals grown from, indicate the temperature, what device was used to record the crystal structure.
Author Response
Dear Reviewer, we would like to thank you for your valuable review and comments, which – we believe – helped us significantly to improve our manuscript. Our amendments stemming from your requests are listed in the attached Word document.

Round 2
Reviewer 1 Report (Previous Reviewer 3)
Comments and Suggestions for Authors
I think that the auhors have carried out a sufficient work to improve the manuscript.
Reviewer 3 Report (New Reviewer)
Comments and Suggestions for Authors
Dear authors,
let me give you a piece of advice for the future. As soon as you synthesize/purify any compound, you are obliged to characterize it using modern methods. You cite work [5], but this is absolutely unconvincing. You are obliged to characterize the product you are working with. I hope that you will understand and will not make such a mistake in the future. I sincerely wish you success.
This manuscript is a resubmission of an earlier submission. The following is a list of the peer review reports and author responses from that submission.
Round 1
Reviewer 1 Report
Comments and Suggestions for Authors
This manuscript titled “promising protocol for in vivo experiments with betulin”, however, it lacks significant novelty, which limits its impact. Moreover, authors should highlight the unique aspects of their protocol compared to existing studies. This manuscript does not provide sufficient details on the betulin extraction method, including the method validation, extraction recovery in the plasma and tissues, any matrix effect observed.
1. The external standard method was used for quantification instead of an internal standard. For a robust and reliable quantification protocol, the internal standard method is generally preferred and should be considered.
2. The manuscript suggests that LC-MS would be a better technique for quantification, yet GC-MS was chosen. The authors should justify this decision with supporting data or literature references.
3. There is no discussion of key method validation parameters such as extraction recovery, matrix effect, accuracy, and precision. These should be included to demonstrate the reliability of the analytical method.
4. In Section 2.1.3, there is a repeated paragraph. Please revise to ensure clarity and avoid redundancy.
5. The plasma PK profile is missing and should be included to strengthen the pharmacokinetic analysis.
6. The manuscript should include detailed PK parameters and tissue distribution data in tumor-bearing mice under combination treatment conditions.
7. The tissue distribution study was conducted using only two rats in Figure 3, which is not statistically acceptable. A larger sample size is necessary to ensure reliable and reproducible results.
Comments on the Quality of English LanguageThe English could be improved to more clearly express the research.
Reviewer 2 Report
Comments and Suggestions for Authors
This study presents a simple way to improve betulin delivery and absorption using oil-based carriers. It also shows betulin’s potential synergy with Namitecan for cancer treatment. The design is good; however, more research is needed to improve the manuscript, specifically on stability, pharmacokinetics, and safety.
1. The stability assessment of the betulin formulation is limited. The study mentions that solutions were stable over one month at 4°C, but no long-term stability data is provided. Additional studies at different temperatures (e.g., room temperature and accelerated stability conditions) are necessary.
2. The study lacks detailed chemical stability analysis under varying pH and light exposure conditions. These factors could also influence betulin degradation, particularly in clinical applications.
3. There is no mention of microbial stability or potential contamination risks, which are essential for formulations intended for long-term storage.
4. There is no comparison with previously established nanoformulations, such as nanoemulsions or liposomal formulations, which have been widely studied for poorly soluble compounds like betulin.
5. The study lacks pharmacokinetic analysis. Such data would provide a clearer view of betulin’s absorption and elimination profile.
6. There is no evaluation of formulation toxicity. Although the study reports betulin’s effects on Ehrlich adenocarcinoma, potential off-target toxicity, organ function impact, and histopathological assessments were not well conducted.
7. The synergy between betulin and Namitecan was observed but not mechanistically explained. Further studies on molecular pathways involved in this interaction would strengthen the findings.
Reviewer 3 Report
Comments and Suggestions for Authors
Please see the attachment.
